# Clinical Characteristics in the Longitudinal Follow-Up of APECED Syndrome in Southern Croatia—Case Series

**DOI:** 10.3390/genes13040558

**Published:** 2022-03-22

**Authors:** Veselin Skrabic, Ivna Skrabic, Roko Skrabic, Blanka Roje, Marko Simunovic

**Affiliations:** 1Department of Pediatrics, University Hospital of Split, Spinciceva 1, 21000 Split, Croatia; ivna595@gmail.com (I.S.); markosimunovic@hotmail.com (M.S.); 2Department of Pediatrics, University of Split School of Medicine, Soltanska 2, 21000 Split, Croatia; 3Department of Nephrology, University Hospital of Split, Spinciceva 1, 21000 Split, Croatia; roko.skrabic@gmail.com; 4Laboratory for Cancer Research, University of Split School of Medicine, Soltanska 2, 21000 Split, Croatia; borjeism@gmail.com

**Keywords:** APECED, APS1, hypoparathyroidism, Addison’s disease, chronic mucocutaneous candidiasis

## Abstract

Background: Autoimmune polyendocrinopathy candidiasis ectodermal dystrophy (APECED) is a rare monogenetic autosomal recessive disorder caused by a mutation in the autoimmune regulator (*AIRE*) gene characterized by complex phenotypic characteristics discovered over years of follow-up. Methods: 7 patients were recruited in this case series in a period of the last 37 years from Southern Croatia. All patients were screened for *AIRE* R257X mutations. Results: This study group had a mean current age of 25.3 years (age range from 5.4 to 40.2 years), while the mean age at the onset of the disease was 6.5 years (age range from 0.7 to 9.2 years) and with a mean follow-up period of 17.8 years. The overall prevalence of APECED syndrome is estimated to be 1 in 75,000. The most common initial manifestation of the disease was onychodystrophy, while the first major component of APECED syndrome was chronic mucocutaneous candidiasis. Conclusions: APECED is a ‘‘multi-faced’’ disease based on the very unpredictable and inconsistent onset of major components. Furthermore, based on our results, we suggest that onychodystrophy could be included as a warning sign of APECED syndrome.

## 1. Introduction

Autoimmune polyendocrinopathy candidiasis ectodermal dystrophy (APECED) (OMIM 240300), also known as autoimmune polyglandular syndrome type 1 (APS1), is a rare monogenetic autosomal recessive disorder caused by a mutation in the autoimmune regulator (*AIRE*) gene characterized by complex phenotypic characteristics discovered over years of follow-up [1,2,3,4]. Today, more than 120 mutations of the *AIRE* gene are reported, which are located on the chromosome 21q22.3 and play an important role in the regulation of the negative selection of auto-reactive T-cells in the thymus [5,6,7]. The prevalence of APECED varies depending on the observed population, but it is well known that the prevalence is higher in populations with a high-level of genetical isolation, such as: Iranian Jews (1 in 9000), Finns (1 in 14,000), and Sardinians (1 in 25,000) [1,8,9]. A clinical diagnosis of APECED is defined by the presence of two out of the three components of the classical triad: hypoparathyroidism, Addison’s disease (AD), and chronic mucocutaneous candidiasis (CMC) [3,10]. In addition, multiple other clinical manifestations occur as part of APECED syndrome, including endocrinopathies due to autoimmune tissue disruption, and ectodermal dysplasia, most commonly a defect in enamel and nails [11,12]. Furthermore, the syndrome includes various organ-specific autoimmune diseases with highly variable prevalence and manifestation time [12,13]. Based on this finding, organ-specific autoantibodies can be found in patients with functional failure as a consequence of the lymphocytic infiltration of the affected tissue, such as autoantibodies on 21-hydroxylase and side-chain-cleavage enzyme [14,15]. Unfortunately, these autoantibodies are not specific diagnostical markers for the APECED because they are specific for target organs and not to the disease itself [15]. However, recently it has been shown that almost all patients with the APECED display autoantibodies to interferons and interleukins, and especially for interferon omega (INFω) and interferon alfa (INFα), which may appear years before the onset of the first clinical symptoms and are possible good diagnostic screening markers [2,14].

The aim of this case series was to determine the prevalence of APECED in the Southern part of Croatia and demonstrate the complex and unpredictable relationship between genotype and phenotype in our group during the long period of follow-up.

## 2. Materials and Methods

The study was approved by the Ethics Committee of the University Hospital of Split and is conducted in accordance with the 1975 Declaration of Helsinki and its later amendments. The study group were recruited in a period of the last 37 years from the Dalmatia, Southern Croatia, and all patients were referred to the Division of Pediatric Endocrinology of the University Hospital in Split. Informed written consent was obtained from the parents or patients of all the study participants.

### 2.1. Clinical Assessment and Follow-Up

All patients received a follow-up at least once a year, and standard visit protocol included clinical examination, anthropometric measurements, and hormonal status. Furthermore, depending on the clinical symptoms, patients were referred to other specialists such as an ophthalmologist, pediatric gastroenterologist, pediatric hematologist, pediatric neurologist, pediatric nephrologist, and dentist.

### 2.2. Genetic Analysis of the AIRE Gene

All patients were screened for *AIRE* R257X mutations. DNA was isolated using Qiazol reagent (Qiagen, Hilden, Germany) according to the manufacturer’s protocol. Approximately 200 ng of DNA was used for PCR reaction. The reaction was performed on a PCR Arktik Thermal Cycler (Thermo Scientific) using the following conditions: initial denaturation step at 95 °C for 10 min, followed by 35 cycles of amplification (95 °C for 1 min, 55 °C for 1 min, 72 °C for 1 min) with final extension step at 72 °C for 5 min. Following primer set was used: 5′-GCGGCTCCAAGAAGTGCATCCAGG-3′ and 5′-CTCCACCCTGCAAGGAAGAGGGGC-3′. Sequences were analyzed using package sangerseq R in R programming language (ver. 3.6.3) [16].

## 3. Results

This study group of seven subjects with APECED syndrome have a mean current age of 25.3 years (age range from 5.4 to 40.2 years), while the mean age at the onset of the disease was 6.5 years (age range from 0.7 to 9.2 years) and with a mean follow-up period of 17.8 years. In addition, four patients were male, three patients were female, and patients number 1 and 2 and patients number 5 and 6 were siblings. The overall prevalence of APECED syndrome in Southern Croatia is estimated to be 1 in 75,000. All patients were homozygous for the R257X (c.769C > T) mutation in the *AIRE* gene (Figure 1).

### 3.1. Initial Clinical Characteristics

The chronological presentation of diseases in all the patients with APECED syndrome is presented in Figure 2.

The most common initial manifestation of the disease was onychodystrophy in four (57.1%) patients (Figure 3), while the first major component of APECED syndrome was CMC in 4 (57.1%) patients and hypoparathyroidism in three (42.9%) patients.

### 3.2. Endocrinological Manifestations

All of our patients developed one of the major endocrinological components of APECED syndrome hypoparathyroidism or AD, while one (14.2%) patient developed five endocrinopathies, one (14.2%) had four endocrinopathies, two (28.7%) had three endocrinopathies, one (14.2%) had two endocrinopathies and two (28.7%) patients developed only one endocrinopathy. The additional clinical and laboratory endocrine characteristics of our subjects with APECED syndrome are presented in Table 1.

Furthermore, patient number 2 died from hypoglycemia due to unregulated type 1 diabetes (T1D). In addition, patient number 4 manifested T1DM after more than 17 years of a positive glutamic acid decarboxylase 65-kDA isoform (GADA-65) and islet cell cytoplasmic autoantibodies (ICA) β cell antibodies. None of the male subjects developed testicular failure, while all of our female subjects developed premature ovarian failure.

### 3.3. Non-Endocrinological Manifestations

The frequency of non-endocrinopathies is also variable. One (14.2%) of our patients had 14 non-endocrinopathies, two (28.7%) had ten non-endocrinopathies, two (28.7%) had six non-endocrinopathies, one (14.2%) had three non-endocrinopathies, and one (14.2%) patient had only two non-endocrinopathies. Onychodystrophy manifested in all of our patients.

### 3.4. Rare Manifestations

Gastrointestinal manifestations occurred in six (85.7%) patients and varied from different types of gastrointestinal dysfunctions to more serious complications such as autoimmune hepatitis (patient 1), liver insufficiency (patient 5), and colon perforation (patient 1). Furthermore, five (71.1%) patients have different types of skin diseases including the most common alopecia and albinism of eyelashes. Patient number 4 manifested autoimmune GAD65 positive encephalitis, which led to a severe and life-threatening condition.

## 4. Discussion

In a longitudinal follow-up study, we reported seven subjects with APECED syndrome with different clinical characteristics and phenotype discrepancy regardless of the homogeneity of the *AIRE* gene mutation in our study population. In addition, the first manifestation of the syndrome in six of our patients was non-endocrinopathy, most commonly onychodystrophy, while the most common first endocrinopathy was Addison’s disease. To the best of our knowledge, this study is the first to report the prevalence of APECED syndrome in Croatia.

In most European countries, there is a high heterogeneity of mutations of the *AIRE* gene, which can vary between different regions within the same country [10,14,17,18,19]. This heterogeneity in the *AIRE* gene mutation has been shown in several studies in the Italian population, which demonstrates that the R203X mutation of *AIRE* gene is most common in southern Italy compared with northeastern Italy, where the most frequent R257X mutation is present [13,15,17]. Contrary to findings, in the Finnish population, which is one of the largest longitudinally follow-up cohorts of APECED patients, they reported markedly homogeneous prevalence of mutations in the *AIRE* gene, and 62 patients out of a total of 80 had the R257X mutation [1,3]. Our findings are consistent with the results of a Finnish study and show a homogeneous genotype in patients with APECED syndrome. Furthermore, the R257X mutation of the *AIRE* gene it is the most common mutation in the population of eastern and central Europe, which indicates the consistency of mutations in our region [10,19]. However, this congruence in patients’ genotypes does not correspond to the different presentations of the APECED syndrome in our population, which indicates a complex relationship between genotype and phenotype in APECED syndrome. These phenotypic discrepancies may be explained by unknown epigenetic, molecular, environmental, and/or unowned third factors which could have an influence on the clinical manifestation and also the progression of the disease.

In addition, onychodystrophy was the first manifestation of APECED syndrome in most of our patients, which is in contrast with other studies. Myhre et al. reported that only one of their patients had onychodystrophy as a first sign of illness and gradually developed in 10% of their study population of patients with APECED syndrome in Norway [20]. In a Finnish study, the most common initial manifestations were components of the classical triad: CMC and hypoparathyroidism, while onychodystrophy was found later in 52% of patients [1,3]. Our findings further emphasize the heterogeneity of the phenotypes of APECED syndrome as a truly multifaced disease, and possibly indicate the importance of onychodystrophy as new waring signs for the APECED syndrome. However, larger studies are needed which will focus on possible unrecognized first manifestations of the disease to further clarify the importance of onychodystrophy as a trigger for the screening for APECED syndrome.

Furthermore, two pairs of siblings from our study showed significant phenotypic variability regardless of the identical *AIRE* mutation, which cannot be completely clarified by the various external triggers associated with the APECED syndrome [21]. Additional total exon sequencing studies are needed to elucidate the wider possible genetic causes of phenotypic variations.

In comparison with other neighboring populations, we reported a lower prevalence of APECED syndrome, which varies in Italy depending on the observed region from 1 in 14,400 to 1 in 35,000 [9,13,22]. Additionally, in the Slovenian population, prevalence is estimated to be 1 in 43,000, which is slightly lower compared to the Finnish cohort where the prevalence is 1 in 25,000 [1,10,23]. However, we reported higher prevalence compared to the Norwegian population 1 in 800,000, Polish population 1 in 129,000 and Irish population 1 in 130,000 [11,20,24].

This study has some limitations. First, the smaller size of the study group and the size of the study region might influence the generalizability of our result. Second, in our study we did not measure the levels of specific autoantibodies (interferons and interleukins) which may also have affected our results. Finally, the point on genetic analysis of only the R257X mutation and not the overall sequencing of the *AIRE* gene could possibly have an effect on the unknown genetic cause of phenotypic differences in our study group.

## 5. Conclusions

In conclusion, we confirmed that APECED is a ‘‘multi-faced’’ disease based on the very unpredictable and inconsistent onset of major components in our genetically homogeneous cohort of patients. Based on our results, we suggest that onychodystrophy could be included as a warning sign of APECED and a possible trigger for early screening, which might possibly prevent endocrine and non-endocrine life-threatening complications. Future research is still needed to completely clarify the relationship between genotype and phenotype in larger international cohorts with long-term follow-up of patients with APECED.

## Figures and Tables

**Figure 1 genes-13-00558-f001:**
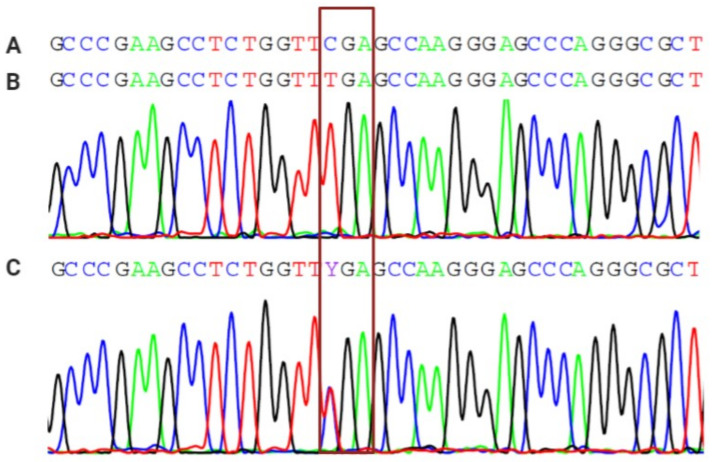
Sequencing analysis for the R257X (c.769C > T) mutation in *AIRE* gene: (**A**) Healthy homozygote. (**B**) Homozygote for R257Xmutation with c.769C > T conversion. (**C**) Heterozygote for R257Xmutation with c.769C > T conversion in one allele.

**Figure 2 genes-13-00558-f002:**
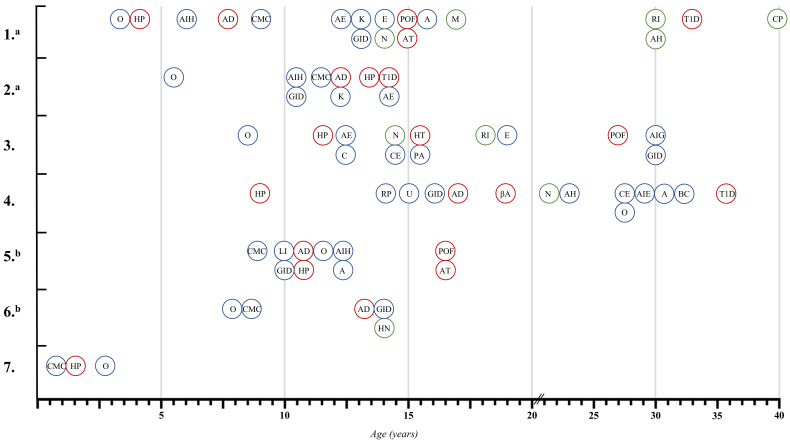
Chronological presentation of diseases in all of patients with APECED syndrome enrolled in the study. O, onychodystrophy; HP, hypoparathyroidism; AIH, autoimmune hepatitis; AD, Addison’s disease; CMC, Chronic mucocutaneous candidiasis; AE, albinism of eyelashes; K, keratitis; GID, gastrointestinal dysfunction; E, enamel dysplasia; N, nephrocalcinosis; POF, premature ovarian failure, AT, autoimmune thyroiditis; A, alopecia; M, megacolon; RI, renal insufficiency; AH, albinism of hair; T1D, type 1 diabetes; CP, colon perforation; C, cheilosis; CE, candidiasis of esophagus; HT, hyperthyroidism; PA, pernicious anemia; AIG, autoimmune gastritis; RP, retinitis pigmentosa; U, prolonged urticaria and angioedema; βA, positive GADA- 65 and ICA β cell antibodies; AH, arterial hypertension; AIE, autoimmune encephalitis; BC, bilateral cataract; LI, liver insufficiency; HN, hydronephrosis; a siblings; b siblings. Endocrinopathies are marked with red circle, while non-endocrinopathies are marked with a blue circle, and non-autoimmune other manifestations are marked with a green circle.

**Figure 3 genes-13-00558-f003:**
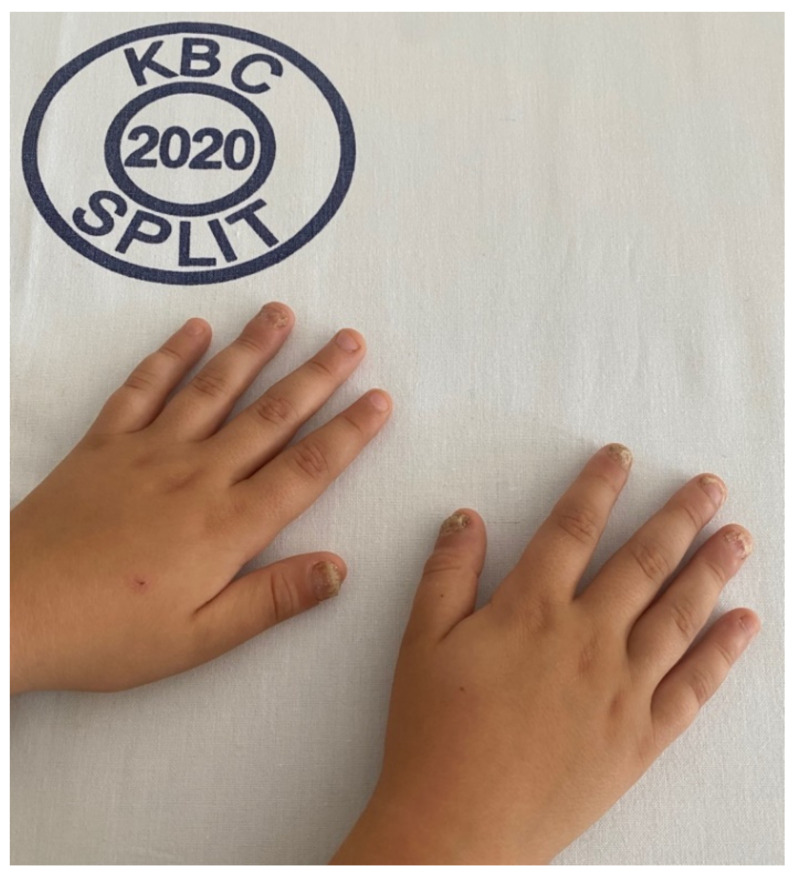
Onychodystrophy.

**Table 1 genes-13-00558-t001:** Endocrinological manifestation and clinical features of the subjects enrolled in the study.

Patient Number	HP	AD	AT	T1D	POF	Laboratory Findings at Diagnosis	Endocrine Complications	Curent Therapy
I	+	+	+	+	+	↓Ca, ↓PTH	N, RI, BGC	H, F, C, I, EP
II	+	+		+		↑AST, ↑ALT, ↑GGT	FH	/
III	+		+		+	↓Ca, ↓PTH	N, RI, BGC	C, EP
IV	+	+		+		↓Ca, ↓PTH	N	H, F, C, I, AE
V	+	+	+		+	↓Gluc, ↑ACTH,↓Cort	/	H, F, C, EP, LT4
VI		+				↑ACTH,↓Cort	/	H, F
VII	+					↓Ca, ↓PTH	/	C

HP, hypoparathyroidism; AD, Addison’s disease; AT, autoimmune thyroiditis; T1D, type 1 diabetes; POF, premature ovarian failure; H, hydrocortisone; F, fludrocortisone; C, calcitriol; EP, estrogen and progesterone; I, insulin; AE, antiepileptic; LT4, levothyroxine; N, nephrocalcinosis; RI, renal insufficiency; BGC, basal ganglia calcification; FH, fatal hypoglycemia; Ca, calcium; PTH, parathyroid hormone; AST, aspartate transaminase; ALT, alanine transaminase; GGT, γ-glutamyl transferase; Gluc, glucose; ACTH, adrenocorticotropic hormone; Cort, cortisol; +: developed; ↑: up; ↓: down.

## Data Availability

The data presented in this study are available on request from the corresponding author. The data are not publicly available due to restrictions e.g., privacy or ethics.

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
