# Peer review of "Clinical Characteristics in the Longitudinal Follow-Up of APECED Syndrome in Southern Croatia—Case Series"

_genes, 2022, doi:10.3390/genes13040558_

Round 1

Reviewer 1 Report

This is a very interesting case series presentation of high scientific and educational value. All my initial remarks has been explained in the limitation section. I would only suggest performing overall sequencing of the AIRE gene in the future analysis. It can provide new data for better understanding of the disease.

I would suggest extensive language correction.

Author Response

Once again we would like to express gratitude for the reviewer comment and we agree as we have stated in the limitations of our manuscript it is necessary to do sequencing of the whole gene in the future studies. 

Reviewer 2 Report

This study describes some clinical characteristics of seven patients with APECED disorder. Although genetics is described in detail, most clinical aspects are missing. A major strength is the long follow-up period.

The authors should include a table detailing labs, current treatments and complications associated with each disease described in Table 1. Did anyone with hypoparathyroidism, for example, have basal ganglia calcification? Or Nephrocalcinosis? What were the labs at diagnosis?

I would enlarge Figure 2 as the circles are too tiny.

Author Response

We would like to address the following comments of the Reviewer 2:

  1. We would like to address the following comment:

“This study describes some clinical characteristics of seven patients with APECED disorder. Although genetics is described in detail, most clinical aspects are missing. A major strength is the long follow-up period. The authors should include a table detailing labs, current treatments and complications associated with each disease described in Table 1. Did anyone with hypoparathyroidism, for example, have basal ganglia calcification? Or Nephrocalcinosis? What were the labs at diagnosis?’’

Comment: According to the Reviewer’s suggestions, we improved the Results of our study to further clarify the clinical features of our patients. We have completely redesigned Table 1 and now it contains data on laboratory findings at diagnosis, endocrine complications and current therapy of our subjects enrolled in the study.

  1. We would like to address the following comment:

”I would enlarge Figure 2 as the circles are too tiny.”

Comment: According to the Reviewer’s suggestions we further enlarge Figure 2.

Round 2

Reviewer 2 Report

The authors have addressed my concerns consistently.